# Dynamic top-down biasing implements rapid adaptive changes to individual movements

**Lucas Y Tian\*†, Timothy L Warren‡, William H Mehaffey, Michael S Brainard\***

Center for Integrative Neuroscience and Howard Hughes Medical Institute, University of California, San Francisco, San Francisco, United States

**\*For correspondence:**
lucasytian@gmail.com (LYT);
msb@phy.ucsf.edu (MSB)

**Present address:** †Laboratory of Neural Systems, Rockefeller University, New York, United States; ‡Departments of Horticulture and Integrative Biology, Oregon State University, Corvallis, United States

**Competing interest:** The authors declare that no competing interests exist.

**Abstract** Complex behaviors depend on the coordinated activity of neural ensembles in interconnected brain areas. The behavioral function of such coordination, often measured as co-fluctuations in neural activity across areas, is poorly understood. One hypothesis is that rapidly varying co-fluctuations may be a signature of moment-by-moment task-relevant influences of one area on another. We tested this possibility for error-corrective adaptation of birdsong, a form of motor learning which has been hypothesized to depend on the top-down influence of a higher-order area, LMAN (lateral magnocellular nucleus of the anterior nidopallium), in shaping moment-by-moment output from a primary motor area, RA (robust nucleus of the arcopallium). In paired recordings of LMAN and RA in singing birds, we discovered a neural signature of a top-down influence of LMAN on RA, quantified as an LMAN-leading co-fluctuation in activity between these areas. During learning, this co-fluctuation strengthened in a premotor temporal window linked to the specific movement, sequential context, and acoustic modification associated with learning. Moreover, transient perturbation of LMAN activity specifically within this premotor window caused rapid occlusion of pitch modifications, consistent with LMAN conveying a temporally localized motor-biasing signal. Combined, our results reveal a dynamic top-down influence of LMAN on RA that varies on the rapid timescale of individual movements and is flexibly linked to contexts associated with learning. This finding indicates that inter-area co-fluctuations can be a signature of dynamic top-down influences that support complex behavior and its adaptation.

## Editor's evaluation

This paper shows that the output of a songbird basal ganglia cortical loop exhibits activity that significantly covaries with a downstream primary motor area with the temporal specificity necessary to promote learning and adaptive premotor bias. As the first study to record from two distant sites at once in singing birds, this study also provides exceptional evidence for temporally precise coordination between two motor areas in the service of vocal learning. The paper provides fundamental implications for understanding the precise neural mechanisms associated with top-down modification of neural commands during motor learning.

## Introduction

Complex behaviors depend on the coordinated activity of neural ensembles in distinct brain areas, yet much remains to be resolved about the specific behavioral functions of such multi-area coordination. Recent studies of trial-by-trial activity co-fluctuations between neuronal populations in connected brain areas have noted a link between changes to such coordinated activity and behavioral performance (*Cowley et al., 2020*; *Koralek et al., 2013*; *Koralek et al., 2012*; *Lemke et al., 2019*;

*Makino et al., 2017*; *Perich et al., 2018*; *Sawada et al., 2015*; *Veuthey et al., 2020*; *Wagner et al., 2019*). Although some slower activity co-fluctuations may reflect task-independent, shared changes in a global internal state (*Cowley et al., 2020*), more dynamic co-fluctuations—varying rapidly on a moment-by-moment basis and linked flexibly to specific behavioral contexts—could be a signature of task-relevant influences of one area on another (*Semedo et al., 2019*).

One intriguing role for such dynamic inter-area influences has been hypothesized in the context of the learning and adaptation of skilled reaching behavior (*Perich et al., 2018*; *Veuthey et al., 2020*). Specifically, the joint evolution of inter-area activity co-fluctuations and behavioral performance over the course of learning has led to the suggestion that top-down inputs from higher-order areas dynamically implement learning-related changes in activity in primary motor areas. In principle, such top-down modulation during learning could operate to control adaptive behavioral modifications on the fine timescale of individual movements. However, the difficulty in identifying and quantifying discrete movement components during skilled reaching leaves unresolved whether such top-down influences operate on this rapid timescale during learning.

Here, we take advantage of error corrective learning of adult birdsong—a complex learned behavior with quantifiable, rapidly varying discrete movements—to investigate whether inter-area activity co-fluctuations change during learning with the appropriate specificity to constitute such dynamic top-down commands. Previous work has led to the hypothesis that such error corrective learning depends on the moment-by-moment shaping of activity in primary motor areas by dynamic, top-down control. In particular, pharmacologically inactivating either the song-specific higher-order area LMAN or its synapses in the primary motor cortical analog RA (*Figure 1A*) transiently eliminates recently learned modifications to pitch (fundamental frequency, FF) of the individual 50–200 ms syllables that constitute adult song (*Figure 1B*, *Andalman and Fee, 2009*; *Charlesworth et al., 2012*; *Tian and Brainard, 2017*; *Warren et al., 2011*). This raises the possibility that LMAN provides top-down drive to RA to implement adaptive changes to behavior (*Ali et al., 2013*; *Andalman and Fee, 2009*; *Aronov et al., 2008*; *Bottjer et al., 1984*; *Charlesworth et al., 2012*; *Doya and Sejnowski, 1998*; *Fee and Goldberg, 2011*; *Kao et al., 2005*; *Kearney et al., 2019*; *Nordeen and Nordeen, 2010*; *Olveczky et al., 2005*; *Scharff and Nottebohm, 1991*; *Tian and Brainard, 2017*; *Troyer and Bottjer, 2001*; *Troyer and Doupe, 2000a*; *Warren et al., 2011*). Moreover, studies in sleeping and anesthetized birds have identified co-fluctuations in LMAN and RA activity (*Hahnloser et al., 2006*; *Kimpo et al., 2003*)—measured as increases in the cross-covariance between LMAN and RA activity—which peak at a short LMAN-leading temporal lag, consistent with the possibility of a dynamic top-down influence of LMAN on RA. However, it remains unclear whether such LMAN–RA co-fluctuations are present during singing and, if so, whether they change during learning in a manner that could support a top-down role of LMAN in the adaptive adjustment of song parameters.

## Results

### LMAN-leading co-fluctuations of LMAN and RA activity are present during singing

To determine whether the co-fluctuations in neural activity between LMAN and RA previously identified in sleeping and anaesthetized birds are present during singing, we first examined neural activity in both nuclei during baseline singing before the onset of training (*Figure 1A–D*). Consistent with prior studies in which LMAN and RA were recorded in separate birds (*Aronov et al., 2008*; *Chi and Margoliash, 2001*; *Hessler and Doupe, 1999*; *Kao et al., 2008*; *Leonardo and Fee, 2005*; *Yu and Margoliash, 1996*; *McCasland, 1987*; *Olveczky et al., 2005*; *Sober et al., 2008*), average activity in both areas exhibited consistent temporal structure aligned to ongoing song, peaking within 50 ms prior to syllable onsets (e.g., *Figure 1D* and *Figure 1—figure supplement 1A–D*). Additionally, we observed a close alignment between the patterns of activity in the two areas (*Figure 1—figure supplement 1C–F*). Despite such consistent temporal structure in average activity, we observed rendition-by-rendition variation in the patterning of activity in both nuclei, raising the possibility that this variation reflects co-fluctuations in LMAN and RA activity.

We assessed co-fluctuations of LMAN and RA activity by measuring cross-covariance, a metric previously used in sleeping and anaesthetized birds (*Kimpo et al., 2003*; see Methods). For a given pair of LMAN and RA recordings (example in *Figure 1C–E*), LMAN–RA cross-covariance is a measure

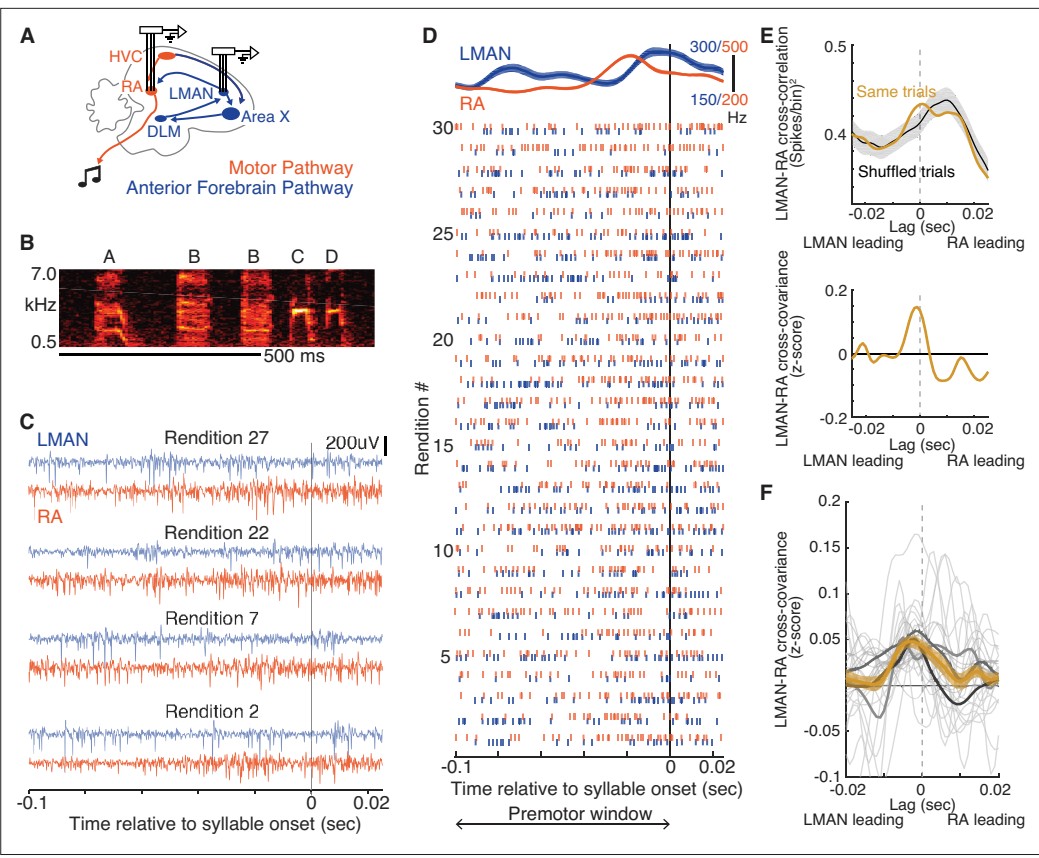

**Figure 1.** LMAN-leading co-fluctuations of LMAN and RA activity are present during singing. (**A**) Schematic of song system circuitry implicated in the production and learning of birdsong. This study focuses on hypothesized top-down signals from LMAN (lateral magnocellular nucleus of the anterior nidopallium), the output nucleus of the Anterior Forebrain Pathway (blue), to RA (robust nucleus of the arcopallium) in the Motor Pathway (red). RA sends projections to brainstem motor nuclei. Recordings of multi-unit activity were made using multi-electrode arrays chronically implanted in LMAN and RA. DLM, medial dorsolateral nucleus of thalamus. HVC and Area X are used as proper names. (**B**) Spectrogram of a single bout of a motif 'ABBCD' A motif is a specific sequence of syllables that is consistently sung across bouts and is unique to a bird (see Methods). Scale bar, 500 ms. (**C**) Example paired recordings in LMAN and RA across four renditions (numbered to match those in panel D) aligned to the onset of a single syllable, filtered in the spike band (300–3000 Hz). (**D**) Raster plot representing spikes for the same pair of LMAN and RA sites shown in panel C, across 30 renditions, ordered chronologically and aligned to syllable onset. Above the raster plot are the mean (± SEM) smoothed firing rates. 'Premotor window' refers to the time window of neural activity used for calculating the cross-covariance between LMAN and RA spike trains. (**E**) Calculation of normalized cross-covariance for the example sites and syllable in panels C and D. Top: cross-correlation between LMAN and RA was calculated using spike trains for both concurrent trials (Same trial) and a control dataset in which LMAN and RA activity patterns were shuffled between adjacent renditions ('Shuffled trials', mean ± standard error of the mean [SEM]). We used this shuffling procedure to estimate the cross-correlation between the mean activity patterns in LMAN and RA (i.e., eliminating the contribution of rendition-by-rendition variation that is shared between LMAN and RA, see Methods). Bottom: the mean of the shuffle-computed cross-correlation functions (Shuffled trials) was subtracted from the actual cross-correlation function (Same trials) to compute cross-covariance. We then divided the cross-covariance by the standard deviation of the shuffled cross-correlations to compute a normalized cross-covariance (measured in z-score). (**F**) LMAN–RA cross-covariance across all syllables and pairs of LMAN and RA sites during baseline singing. The light gray curves represent individual syllables (N = 27), averaged over all simultaneously recorded pairs of LMAN and RA sites (N = 38 total LMAN–RA site pairs, 3 birds). Mean ± SEM cross-covariance (across syllables, with each syllable contributing its mean over recording site pairs) is shown for individual birds (dark gray) and all data (orange).

The online version of this article includes the following figure supplement(s) for figure 1:

**Figure supplement 1.** LMAN and RA average activity patterns align to each other with time lag close to zero.

**Figure supplement 2.** Histological verification of electrode locations.

of the similarity of moment-by-moment fluctuations of LMAN and RA spike patterns away from their respective across-rendition means, computed as a function of the time lag between these patterns (*Perkel et al., 1967*). We computed the cross-covariance within a premotor window for each syllable, which we defined as a 100ms period centered 50 ms preceding syllable onset (*Figure 1D*), chosen based on empirical estimates of when LMAN and RA activity patterns maximally influence acoustic structure of the upcoming syllable (*Fee et al., 2004*; *Giret et al., 2014*; *Kao et al., 2005*; *Kojima et al., 2018*; *Yu and Margoliash, 1996*; *Sober et al., 2008*). In birds singing spontaneously at baseline ('undirected' singing produced in isolation), the LMAN–RA cross-covariance was on average positive during syllable premotor windows (example pair of sites in *Figure 1E*, cross-covariance computed by normalizing data to shuffled controls; summary across pairs of recordings and syllables in *Figure 1F*), similar to cross-covariance measurements in anesthetized and sleeping birds (*Hahnloser et al., 2006*; *Kimpo et al., 2003*). Although there was variation in the magnitude and time lag of cross-covariance peaks across syllables and recording sites (*Figure 1F*, light gray curves) there was a dominant positive peak in the average cross-covariance with LMAN leading at a short time lag (~3 ms, *Figure 1F*; *Kimpo et al., 2003*). This putative influence of LMAN on RA is consistent with the known presence of a direct excitatory projection from LMAN to RA (*Bottjer et al., 1989*; *Kubota and Saito, 1991*; *Mooney and Konishi, 1991*; *Nottebohm et al., 1982*) and a hypothesized role of LMAN in driving motor variability in RA during baseline singing (*Giret et al., 2014*; *Kao et al., 2005*; *Kao and Brainard, 2006*; *Kojima et al., 2018*; *Olveczky et al., 2005*; *Stepanek and Doupe, 2010*). This indication of LMAN–RA inter-actions during baseline singing raises the question of whether similar top-down signals from LMAN to RA operate during learning to instantiate rapid adaptive adjustments to individual syllables.

## LMAN–RA co-fluctuations are enhanced during learning

To determine whether adaptive biasing of pitch during learning is accompanied by changes to co-fluctuations in LMAN and RA activity, we examined LMAN–RA cross-covariance while birds learned modifications to the pitch of individual 'targeted' song syllables in response to pitch-contingent white noise (WN) feedback (*Figure 2A, B*, see Methods; *Ali et al., 2013*; *Andalman and Fee, 2009*; *Charlesworth et al., 2012*, *Charlesworth et al., 2011*; *Tian and Brainard, 2017*; *Tumer and Brainard, 2007*; *Warren et al., 2011*). Across experiments, we found that LMAN–RA cross-covariance was enhanced following training (3–10 hr) with pitch-contingent reinforcement (*Figure 2C–F*, 'Target syllable'; also *Figure 2—figure supplement 1A*) and this enhancement was maximal at a short LMAN-leading time lag (~2 ms, *Figure 2D*). Moreover, LMAN–RA cross-covariance increased with a similar time course to pitch modification (*Figure 2G, H*). In contrast, there were no detectable changes in LMAN–RA cross-covariance for non-targeted syllables (*Figure 2E, F*, 'Non-target syllables'), including those directly preceding the target syllable (*Figure 2—figure supplement 1B*). There was also no detectable learning-related difference between target and non-target syllables in changes in the overall level of activity within each nucleus (*Figure 2—figure supplement 1C*). The observations that increases in LMAN–RA cross-covariance are restricted to the target syllable and develop in parallel with pitch modification indicate that these changes are linked specifically to learning, and not to other processes such as arousal, non-stationarity or 'drift' of recorded units, or circadian variation in activity patterns.

## The strength of LMAN–RA co-fluctuations and the strength of adaptive motor bias are associated on a rendition-by-rendition basis

If changes to LMAN–RA cross-covariance reflect top-down commands that adaptively bias RA activity and pitch during learning, then rendition-by-rendition variation in the strength of this signal should be associated with rendition-by-rendition variation in the magnitude of pitch shifts in behavior. To test this prediction, we took advantage of the rendition-by-rendition variation in pitch shifts that birds naturally exhibit (*Figure 3A*, note that this variation is reflected in the large spread of pitch values in both 'baseline' and 'trained' periods). We divided interleaved syllable renditions into two groups, based on whether each rendition's pitch was shifted in the adaptive direction by more or less than the median amount (*Figure 3A*). At the peak of learning ('trained' period) we found that renditions exhibiting stronger learning-related pitch shifts (stronger bias) also exhibited greater learning-related enhancement of cross-covariance (example experiment in *Figure 3B*; summary in *Figure 3C*; also see *Figure 3—figure supplement 1A*), regardless of whether birds had been trained to adaptively shift pitch up or down (*Figure 3—figure supplement 1B*). This finding that pitch modifications and

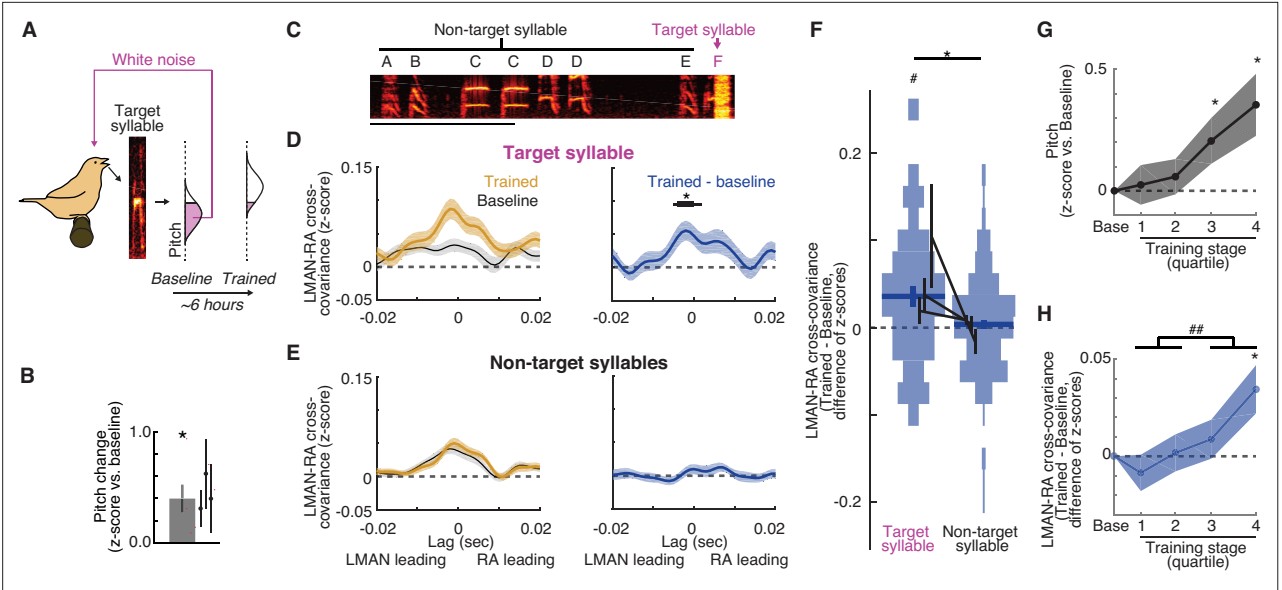

**Figure 2.** LMAN–RA co-fluctuations are enhanced during learning. (**A**) Pitch training paradigm. White noise (WN) feedback is delivered during renditions of a specific target syllable when its pitch is either below or above a threshold (pink fill in 'pitch' histogram), depending on whether the objective is to train pitch shifts up or down, respectively. The schematic shows training for upward pitch shifts. Over the course of training (3–10 hr, mean ~6 hr), birds progressively modify pitch in the direction that escapes WN so that in this example the 'Trained' distribution is shifted upwards relative to 'Baseline'. (**B**) Summary of the magnitude of pitch change across experiments (mean ± standard error of the mean [SEM], $N = 11$ experimental trajectories over 3 birds); individual points represent the mean for individual birds. *$p < 0.05$, Wilcoxon signed-rank test. (**C**) Spectrogram of a song bout in an example experiment, with the syllable 'F' targeted with pitch-contingent WN. Scale bar, 500 ms; y-axis, 0.5–7.0 kHz. (**D**) Change in cross-covariance during learning for target syllables. Left: mean ± SEM cross-covariance for 'Baseline' and 'Trained' periods (last quarter of renditions during the training session) across experiments ($N = 38$ LMAN–RA site pairs, 11 experiments, 3 birds). Right: mean ± SEM change in cross-covariance over the course of training. Black bar indicates time bins with values significantly different from zero (thin, $p < 0.05$; thick, $p < 0.005$, Wilcoxon signed-rank test). (**E**) Same as (**D**), but for non-target syllables. For a given experiment, there was only one target syllable, but multiple non-target syllables (mean, 4.2 syllables). Thus for each LMAN–RA site pair, data were first averaged across all non-target syllables before plotting ($N = 38$ pairs, 11 experiments, 3 birds). (**F**) Summary of change in LMAN–RA cross-covariance during training for target and non-target syllables. For each combination of paired LMAN–RA sites and syllables, we computed the average change in cross-covariance (Trained – Baseline) in a 15-ms window centered at the peak of the average end-of-training cross-covariance (−3 ms) [$N = 38$ (Target) and 158 (Non-target) combinations of paired sites and syllables, across 11 experiments in 3 birds]. *$p < 0.05$, mixed effects model (fixed intercept and effect of syllable type; random effect of intercept and syllable type grouped by experiment ID). #$p < 0.05$, mixed effects model (fixed intercept and random effect of intercept grouped by experiment ID). (**G**) Time course of pitch change. Each training trajectory was analyzed by binning renditions into four training stages with equal numbers of renditions (i.e., quartiles). The average pitch change across experiments is plotted for each training stage ($N = 10$ experiments in 3 birds; excluding one experiment for which neural data was recorded only during baseline and the end of training.). Spacing of stages along the x-axis maintains the relative timing of stages (time of median rendition for each stage relative to median baseline rendition: stages 1, 2, 3, and 4: 1.02, 2.35, 3.73, and 5.46 hr). *$p < 0.05$ vs. 0, Wilcoxon signed-rank test. (**H**) Time course of change in LMAN–RA cross-covariance for the target syllable for the same experiments illustrated in (**G**) ($N = 37$ LMAN–RA site pairs, 10 experiments, 3 birds). *$p < 0.05$ vs. 0, Wilcoxon signed-rank test; ##$p < 0.005$, last two vs. first two training quartiles.

The online version of this article includes the following figure supplement(s) for figure 2:

**Figure supplement 1.** Specificity of enhancement of LMAN–RA co-fluctuations during learning.

**Figure supplement 2.** Lack of detected relationship between baseline activity-pitch correlations and learning-related changes in LMAN–RA cross-covariance.

LMAN–RA cross-covariance are linked on a rendition-by-rendition basis suggests that LMAN–RA co-fluctuations reflect a rapidly varying top-down signal that adaptively biases motor performance during learning.

## Learning-related increases in LMAN–RA co-fluctuations are context specific

Prior studies revealed that pitch modifications for a given syllable are dependent on sequential context, or the sequence of syllables in which it is embedded (*Hoffmann and Sober, 2014*; *Tian and Brainard, 2017*); this prompted us to examine whether increases in LMAN–RA cross-covariance are

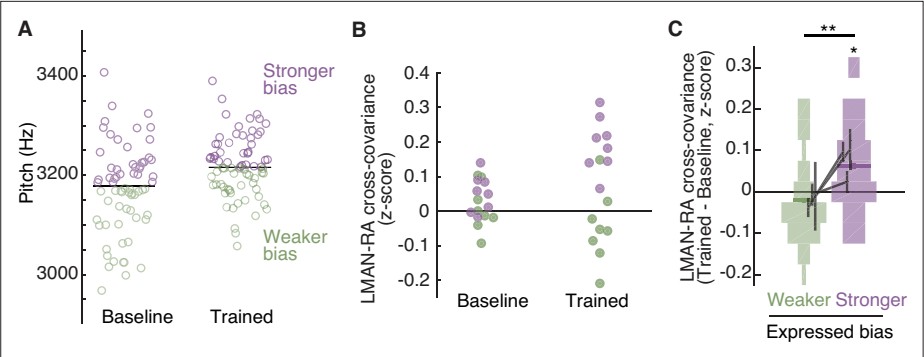

**Figure 3.** The strength of LMAN–RA co-fluctuations and the strength of adaptive motor bias are associated on a rendition-by-rendition basis. (**A**) Example experiment, plotting pitch of individual renditions during baseline and at the end of training in an experiment in which white noise (WN) feedback targeted lower pitch renditions (i.e., training upwards pitch shift). Renditions are split into two groups, 'Stronger bias' (purple) and 'Weaker bias' (green) based on pitch deviation from the median. Renditions in the 'stronger' and 'weaker' groups are interleaved in time, and taken from periods with relatively stable pitch, so that any differences in LMAN–RA cross-covariance between these two groups reflects rendition-by-rendition variation, not slower 'drift' due to learning. We used the last half of baseline and training renditions. Each data point represents a single syllable rendition. (**B**) Example experiment (same as in panel A), plotting LMAN–RA cross-covariance for individual pairs of LMAN–RA sites. Each data point represents the mean cross-covariance for renditions with either stronger bias (purple) or weaker bias (green). Thus, each site pair contributes two points to the 'Baseline' and two points to the 'Trained' data (one purple, one green). The renditions used to calculate cross-covariance are the same as in panel A. Each data point represents a single LMAN–RA site pair. (**C**) Summary of change in LMAN–RA cross-covariance at the peak of training, relative to baseline (Trained − Baseline), measured separately for renditions expressing 'stronger' or 'weaker' bias (*N* = 38 LMAN–RA site pairs, 11 experiments, 3 birds). *p < 0.05, LMAN–RA cross-covariance (Trained − Baseline) modeled with fixed intercept and random intercept grouped by experiment ID. **p < 0.005, Stronger–Weaker modeled with fixed intercept and random intercept grouped by experiment ID.

The online version of this article includes the following figure supplement(s) for figure 3:

**Figure supplement 1.** Robustness of the rendition-by-rendition relationship between LMAN–RA co-fluctuations and adaptive motor bias.

similarly context specific. To do so, we took advantage of the specificity with which learning can be driven; when a syllable (e.g., B) is targeted in a specific sequential context (e.g., AB) and not in others (e.g., XB or YB), the resulting pitch modification is greatest in the targeted context (*Hoffmann and Sober, 2014*; *Tian and Brainard, 2017*). In a subset of experiments, we therefore provided WN feedback in such a context-specific manner, with target and non-target contexts naturally interleaved from rendition to rendition (*Figure 4A, B*). We found that average LMAN–RA cross-covariance increased over the course of training specifically for the target context, with no detected change for the same syllable in non-target contexts (*Figure 4C*; see *Figure 4—figure supplement 1*). Enhanced LMAN–RA cross-covariance is therefore flexibly linked to sequential contexts associated with learning.

## Adaptive bias is eliminated by disrupting LMAN activity in a narrow premotor window

The close relationship between changes to LMAN–RA cross-covariance and behavior is consistent with a model in which LMAN provides a temporally localized top-down command that has a transient, adaptive influence on the immediately upcoming syllable. This model has remained untested in prior studies driving learning and then perturbing LMAN activity to causally probe its contributions to pitch shifts, as these studies have used pharmacological manipulations that act on the timescale of minutes, and thus lack the temporal resolution to assess the moment-by-moment relationship between neural activity and behavior (*Andalman and Fee, 2009*; *Tian and Brainard, 2017*; *Warren et al., 2011*). We reasoned that electrical microstimulation of LMAN could potentially enable a more temporally precise disruption of neural activity and thus test of LMAN's causal influence on RA. Stimulation of LMAN (*Giret et al., 2014*; *Kao et al., 2005*; *Kojima et al., 2018*) and interconnected song system nuclei (*Ashmore et al., 2005*; *Fee et al., 2004*; *Vu et al., 1994*) during baseline singing perturbs song

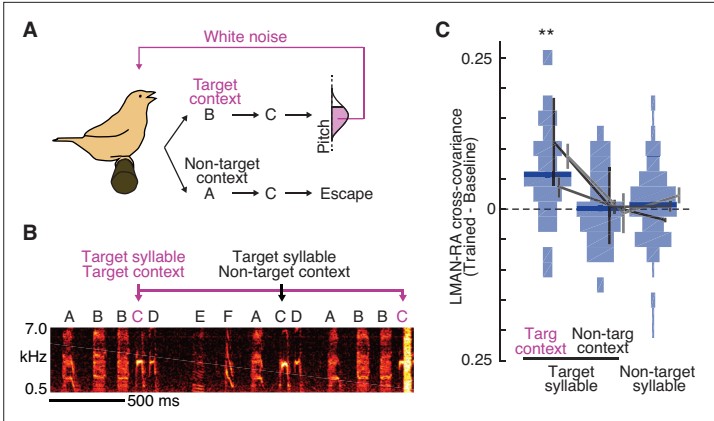

**Figure 4.** Learning-related increases in LMAN–RA co-fluctuations are context specific. (**A**) Schematic of context-dependent training. In this example experiment, pitch-contingent white noise (WN) was provided for renditions of the target syllable (**C**) only in the 'Target' context (BC). WN was never provided when C was sung in 'Non-target' contexts (e.g., AC). (**B**) Spectrogram for the same experiment as in panel A, illustrating pitch-contingent WN delivered only for syllable C in the target context BC. The first rendition of BC escaped WN because its pitch was higher than the threshold defined for this experiment. (**C**) Summary of change in LMAN–RA cross-covariance during training (N = 24 LMAN–RA site pairs for target syllable, target context, 35 pairs for target syllable, non-target context, and 112 pairs for non-target syllables, 7 experiments, 3 birds). **p < 0.005, mixed effects model (fixed effect of intercept and syllable type; random effect of intercept and syllable type grouped by experiment ID).

The online version of this article includes the following figure supplement(s) for figure 4:

**Figure supplement 1.** Learning-related increases in LMAN–RA co-fluctuations are context specific.

acoustic features in a manner that is rapid and transient (timescale of 10 s of milliseconds). LMAN microstimulation—with intensity titrated to minimize overt behavioral effects during baseline singing (see Methods)—thus has the potential to disrupt neural activity that may convey a motor-biasing signal during learning.

To evaluate the specificity of the top-down bias reflected in increased LMAN–RA cross-covariance, we stimulated LMAN within the target syllable's premotor window during a randomly interleaved 50% of renditions ('Stim'; see Methods) in 1- to 3-hr long blocks (*Figure 5A, B*). Compared to prior studies using spatially localized, unilateral microstimulation of LMAN (*Giret et al., 2014*; *Kao et al., 2005*), our stimulation experiments were designed to cause more global, bilateral disruption of LMAN (see Methods). We measured the effect of stimulation by comparing stimulated renditions to randomly interleaved control renditions for which stimulation was withheld. Before the onset of pitch-contingent WN training, stimulation caused modest changes to pitch that varied across experiments, resulting in only small changes to average pitch (example experiment in *Figure 5C, D*; summary in *Figure 5E*; see also *Giret et al., 2014*; *Kao et al., 2005*; *Kojima et al., 2018*). In contrast, during learning, stimulation caused pitch to revert systematically toward its baseline value (*Figure 5C–E*). This reversion is consistent with the interpretation that stimulation removes an adaptive biasing signal that develops during learning, with the remaining expressed pitch shifts reflecting learning that has transferred to the downstream motor pathway to become independent of LMAN (*Andalman and Fee, 2009*; *Warren et al., 2011*). Indeed, this occlusion of pitch modification via temporally precise disruption of LMAN activity was quantitatively indistinguishable from that caused by longer lasting pharmacological inactivation of LMAN in the same species and training paradigm (*Warren et al., 2011*; compare *Figure 5E, F*). Thus, our findings indicate that appropriately patterned LMAN premotor activity is required for adaptive biasing of vocal output during learning.

To precisely localize the temporal window in which LMAN contributes to adaptively biasing motor output, we systematically varied the timing of stimulation relative to the target syllable, applying short stimulation trains (10 ms) at times ranging from −65 to +5 ms relative to the timepoint of maximal pitch modification (*Figure 5G, H*, average time of WN delivery labeled 'WN time'; *Charlesworth et al., 2011*). Stimulation caused reversion of learning only if applied within a narrow premotor window prior to pitch measurement (e.g., 25–45 ms before pitch measurement for the example experiment

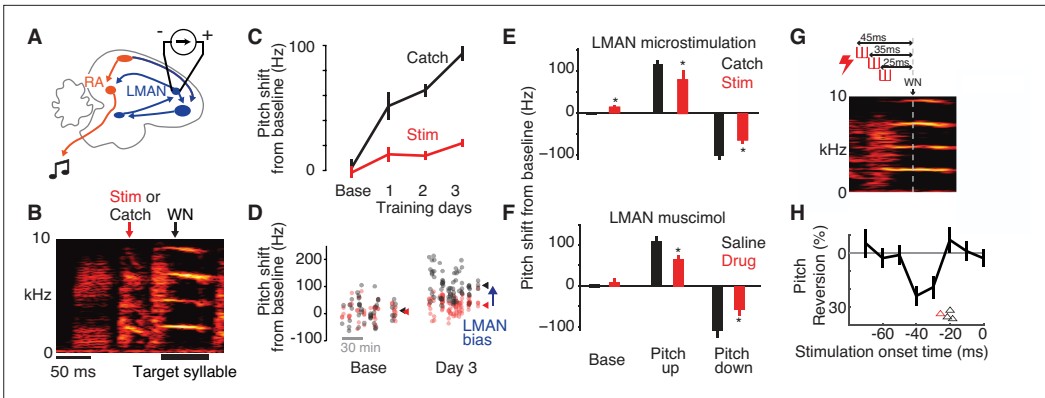

**Figure 5.** Adaptive bias is eliminated by disrupting LMAN activity in a narrow premotor window. (**A**) Schematic of electrical microstimulation in LMAN. Stimulation was used to disrupt LMAN activity at precise times during singing. (**B**) Experimental design. To test for a causal contribution of LMAN premotor activity to pitch modifications during learning, on randomly interleaved renditions stimulation was either delivered ('Stim', 60 ms duration centered 50 ms prior to syllable onset) or withheld (Catch). 'WN' marks the average time of WN feedback for this experiment. (**C**) An example experiment in which pitch was shifted away from baseline over 3 days, with LMAN microstimulation performed during 1–3 hr blocks on the baseline day and 3 subsequent training days, depicting pitch (mean ± standard error of the mean [SEM]) for stimulated (Stim) and non-stimulated (Catch) syllable renditions. (**D**) Scatterplot showing pitch of individual renditions from the experiment in panel C on randomly interleaved stimulated (red) and catch (black) renditions during baseline and training day 3 (arrowheads, mean pitch). The magnitude of LMAN bias is estimated as the difference between Stim and Catch renditions (blue arrow). (**E**) Summary of effects of LMAN microstimulation on pitch across all experiments ($N$ = 14 training sessions in 4 birds), plotted for baseline (Base), and training (mean over training days 1–3) for experiments training pitch up or down. *$p < 0.05$, $t$-test comparing Catch to Stim. (**F**) Same as panel E, but for experiments in which LMAN was inactivated pharmacologically using muscimol or lidocaine (Drug). Pharmacological perturbation had quantitatively similar effects on pitch compared to electrical microstimulation in panel E. Data were previously published (*Warren et al., 2011*). (**G**) Schematic of experiment using short-duration microstimulation [red hash marks, durations 10 ms ($n$ = 3 experiments) or 15 ms ($n$ = 1)] applied at varying timepoints in the target syllable's premotor window. Onset timing of stimulation was varied from −65 to +5 ms relative to the expected timepoint of WN delivery (WN). Three example microstimulation trains are shown, with onsets at −45, −35, and −25 ms. (**H**) Analysis of the temporal latency between short-duration microstimulation and pitch reversion. Black trace shows pitch reversion for the example experiment in Panel G, measured for a single 8ms time bin, caused by stimulation at varying times relative to the time of pitch measurement ('Stimulation onset time', time between the centers of the time bins for stimulation and measuring pitch). Arrowheads depict, for each experiment, the minimum latency between stimulation and significant pitch reversion, calculated from continuous pitch contours (red arrowhead refers to the experiment shown in panel G).

in *Figure 5G, H*). Across four experiments, the latency between stimulation and detectable pitch reversion ranged from 18 to 26 ms (*Figure 5H*, arrowheads), consistent with prior estimates of the premotor delay between LMAN activity and song (*Giret et al., 2014*; *Kao et al., 2005*; *Kojima et al., 2018*). Combined with prior findings from pharmacological perturbations within this circuit and of changes to LMAN–RA cross-covariance during learning, these microstimulation experiments indicate that LMAN exerts a moment-by-moment, top-down influence on RA to implement adaptive motor bias.

## Discussion

Using paired recordings from LMAN and RA in singing birds, we identified a neural signature of a top-down influence of LMAN on RA, quantified as a short latency, LMAN-leading peak in the cross-covariance of neural activity. This LMAN–RA cross-covariance peak is present even at baseline prior to learning reinforcement-driven pitch modifications (*Figure 1*), consistent with an ongoing role for LMAN in driving rendition-by-rendition exploratory variation in behavior (*Kao et al., 2005*; *Olveczky et al., 2005*). Strikingly, during learning LMAN–RA cross-covariance strengthens in a premotor window closely linked to the individual movement (syllable, *Figure 2*), rendition-by-rendition variation

in the magnitude of adaptive pitch modifications (*Figure 3*), and sequential context (*Figure 4*) associated with learning. Moreover, temporally localized perturbation of LMAN activity specifically within this premotor window causes rapid and transient occlusion of learned changes to pitch (*Figure 5*). Combined, these results indicate that LMAN enables learning by conveying a dynamic top-down command to RA that varies on the timescale of individual movements and is flexibly linked to contexts associated with learning.

## Adaptive bias through dynamic top-down influence on primary motor area output

Our finding that LMAN–RA cross-covariance is a temporally precise neural correlate of adaptive behavioral modifications suggests a similar interpretation for previous work that identified enhanced inter-area co-fluctuations during learning (*Koralek et al., 2013*; *Koralek et al., 2012*; *Lemke et al., 2019*; *Makino et al., 2017*; *Sawada et al., 2015*; *Veuthey et al., 2020*; *Wagner et al., 2019*). In particular, prior studies left unresolved whether such co-fluctuation signals reflect the moment-by-moment implementation of adaptive movements instead of other slower modulatory processes related to motivation, vigor, attention, motor preparation, or other kinds of global brain state changes (*Cowley et al., 2020*; *Hikosaka et al., 2006*; *Ignashchenkova et al., 2004*; *Jaffe and Brainard, 2020*; *Kawashima et al., 2016*; *Müller et al., 2005*; *Noudoost et al., 2010*; *Sawada et al., 2015*; *Stavisky et al., 2017*). Our results indicate that inter-area co-fluctuations in these and other systems may similarly reflect dynamic top-down shaping of primary motor output in driving behavioral adaptation. Moment-by-moment and context-specific links between measures of inter-area co-fluctuations and behavior, similar to those we found between LMAN–RA cross-covariance and pitch shifts during learning, may be detected most readily for forms of learning that involve modification of specific movement parameters with high temporal (*Kawai et al., 2015*; *Medina et al., 2005*; *Medina et al., 2000*; *Narayanan and Laubach, 2006*; *Rueda-Orozco and Robbe, 2015*) or contextual specificity (*Howard et al., 2012*; *Rochet-Capellan et al., 2012*; *Rochet-Capellan and Ostry, 2011*; *Wainscott et al., 2005*), similar to song learning and adaptation (*Charlesworth et al., 2011*; *Lipkind et al., 2017*; *Ravbar et al., 2012*; *Tchernichovski et al., 2001*; *Tian and Brainard, 2017*; *Tumer and Brainard, 2007*).

## Neural mechanisms underlying an adaptive top-down influence of LMAN on RA

Our finding of learning-related increases in LMAN–RA cross-covariance raises the question of what neural mechanisms instantiate this top-down bias during learning. A prevalent hypothesis is that top-down LMAN bias results from the reinforcement of LMAN activity patterns associated with successful behavioral variants (*Andalman and Fee, 2009*; *Brainard and Doupe, 2000*; *Charlesworth et al., 2011*, *Charlesworth et al., 2012*; *Doya and Sejnowski, 1998*; *Fee and Goldberg, 2011*; *Gadagkar et al., 2016*; *Kao et al., 2005*; *Kearney et al., 2019*; *Singh Alvarado et al., 2021*; *Troyer and Bottjer, 2001*; *Troyer and Doupe, 2000a*; *Tumer and Brainard, 2007*; *Warren et al., 2011*). Such reinforcement could lead to a variety of changes to LMAN activity patterns that could alter the influence of LMAN on RA, including population-wide changes in firing rate, altered correlational structure across LMAN neurons (*Brown and Raman, 2018*; *Darshan et al., 2017*; *Kumar et al., 2010*; *Riehle et al., 1997*; *Woolley et al., 2014*; *Zandvakili and Kohn, 2015*), or more complex or distributed changes to the temporal structure of LMAN neurons' firing patterns (*Kao et al., 2008*; *Kao et al., 2005*; *Kojima et al., 2013*; *Olveczky et al., 2005*; *Palmer et al., 2021*). We did not detect any systematic learning-related increases or decreases in average LMAN firing rates during learning (*Figure 2—figure supplement 1C*). Moreover, we did not find evidence that enhancement in cross-covariance during learning was specific to LMAN and RA sites that exhibited the highest baseline correlation with pitch (*Figure 2—figure supplement 2*). Both negative findings—although requiring corroboration using larger datasets—suggest that any changes to LMAN activity during learning may occur in a heterogeneous and complex fashion across neurons.

That increases in cross-covariance can be observed at arbitrarily selected recording locations in RA and LMAN suggests that the associated changes in these areas are distributed across neurons. At least for RA, the possibility that activity changes are correlated on a rendition-by-rendition basis across neurons is supported both by the known architecture of RA in which the inhibitory inputs to multiple projection neurons are highly correlated (*Miller et al., 2017*; *Spiro et al., 1999*) and by an

understanding that correlated activity of RA projection neurons supports the ability of these neurons to drive downstream structures and influence song (*Sober et al., 2008*).

An alternative model is that top-down bias is implemented by altering the efficacy of LMAN synapses in RA—potentially in the absence of changes within LMAN—either by plasticity at these synapses, or by changes to neuromodulatory inputs to RA that act to alter the gain of an appropriate set of LMAN synapses in RA. Indeed, prior work has supported a role for noradrenergic (*Sheldon et al., 2020*; *Solis and Perkel, 2006*) and cholinergic (*Puzerey et al., 2018*) inputs to RA in modulating song. Overall, distinguishing between possible mechanisms for learning-related changes to LMAN's top-down bias will require elucidation by studies monitoring larger numbers of neurons during learning.

## A functional architecture with distinct substrates for fast and slow learning

Our findings suggest that motor skill learning is generally supported by biasing signals generated by frontal circuits (such as the Anterior Forebrain Pathway that contains LMAN) separate from those that drive the core motor program. An advantageous feature of this hierarchical architecture may be to enable rapid behavioral modifications in early phases of learning via top-down signals that flexibly update in response to changing goals, contexts, and reward contingencies, while allowing for the stable representation of core motor programs in primary motor areas (*Kim and Hikosaka, 2013*; *Miller and Cohen, 2001*; *Tian and Brainard, 2017*).

On the longer timescale over which expertise is developed, such top-down biasing may also supervise slower plasticity in primary motor areas to drive long-lasting updates to core motor programs. Such plasticity mechanisms may operate for both birdsong (*Andalman and Fee, 2009*; *Fee and Goldberg, 2011*; *Moorman et al., 2021*; *Tian and Brainard, 2017*; *Troyer and Doupe, 2000b*; *Warren et al., 2011*) and other sensory-motor skills (*Kawai et al., 2015*; *Knudsen, 1994*; *Turner and Desmurget, 2010*), and are consistent with observations that early and late stages of sensory-motor learning are supported by distinct neural substrates (*Andalman and Fee, 2009*; *Aronov et al., 2008*; *Bottjer et al., 1984*; *Dayan and Cohen, 2011*; *Doyon and Ungerleider, 2002*; *Hikosaka et al., 2002*; *Kleim et al., 2004*; *Nordeen and Nordeen, 2010*; *Scharff and Nottebohm, 1991*; *Shadmehr and Holcomb, 1997*; *Warren et al., 2011*).

## Methods

### Animal subjects

We used adult male Bengalese finches [*Lonchura striata domestica*, *N* = 8 (4 for recordings, 4 for microstimulation)] that were bred in our colony and housed with their parents until at least 60 days of age. During experiments, birds were housed individually in sound-attenuating chambers (Acoustic Systems) on a 14/10 hr light/dark cycle with food and water provided ad libitum. All experiments were performed on 'undirected' song (i.e., with no female present). All procedures were in accordance with protocols approved by the University of California, San Francisco Institutional Animal Care and Use Committee (Approval #: AN185512-02E).

### Pitch training paradigm using closed-loop reinforcement

We used a modified version of EvTaf (*Charlesworth et al., 2012*, *Charlesworth et al., 2011*; *Tian and Brainard, 2017*; *Tumer and Brainard, 2007*; *Warren et al., 2011*), a custom-written Labview program (National Instruments), to monitor song and deliver WN feedback in a closed-loop fashion during training (EvTaf is available from the authors upon request) (*Ali et al., 2013*; *Andalman and Fee, 2009*; *Charlesworth et al., 2012*, *Charlesworth et al., 2011*; *Tian and Brainard, 2017*; *Tumer and Brainard, 2007*; *Warren et al., 2011*). Briefly, song was recorded with an omnidirectional lavalier microphone (Countryman), bandpass filtered between 75 Hz and 10 kHz, and digitized at 32 kHz. To detect a specific segment of song (i.e., within a targeted syllable) for targeted reinforcement, the spectrum of each successive 8ms segment of ongoing song was tested online for a match to a spectral template constructed to discriminate the targeted segment from all other song segments. Successful match was based on threshold crossing of the Euclidean distance between this song segment and template. A match signaled detection of the targeted syllable. FF, or pitch, of the matching segment was compared to an FF threshold (*Tumer and Brainard, 2007*). If FF was below threshold

(in experiments driving upwards shifts in FF), or above threshold (in experiments driving downwards shifts in FF), WN feedback [WN, 40–60 ms at 90–95 dB(A)] was delivered with <1 ms latency. Feedback renditions were termed 'hits'.

For context-dependent training, this paradigm was modified so that pitch-contingent WN was delivered only if the Target syllable was sung in a specific Target context, defined by the identity of the sequence of syllables directly preceding the Target syllable. Sequential context was defined for each rendition of the Target syllable and detected online by extending the spectral template matching algorithm (above) to detect conjunctions of the current and preceding syllables, using methods described in *Tian and Brainard, 2017*.

All training trajectories for neural recordings were performed within a single day of continuous recording of neural and singing data. WN was turned on after sufficient baseline data were collected (15–40 song bouts). The average training duration was 6.4 hr (min, 3.0; max, 10.0) starting from when WN was turned on.

For LMAN microstimulation experiments, the training duration was extended to up to 4 days with the goal of eliciting large shifts in pitch that would allow more robust measurement of behavioral effects of stimulation. This approach was chosen to match that of previous pharmacology experiments (*Warren et al., 2011*) which enabled us to directly compare behavioral effects of these different manipulations.

For all pitch training experiments, we initially set the WN pitch threshold to a hit rate of ~70%. Successful learning leads to a progressive decrease in the hit rate; therefore, the pitch threshold was updated over the course of training to maintain a hit rate of 70%.

## Offline analysis of song

For analysis of song recorded simultaneously with neural recordings, we used song acoustic data recorded using the same Intan acquisition system used for collecting neural data (see below), to ensure temporal alignment of neural and singing data. Audio signals were acquired with an electret microphone (CUI), amplified (MAX4466, Adafruit), and digitized at 30 kHz. For analysis of song during LMAN stimulation, we analyzed data saved using the Labview training program described above.

Syllable pitch was calculated in the following manner (*Charlesworth et al., 2011*). For each syllable rendition, we calculated a spectrogram using a Gaussian-windowed ($\sigma$ = 1 ms) short-time Fourier transform (window size = 1024 samples; overlap = 1020 samples; sampling rate = 30 kHz). Within each time bin, FF was defined as the frequency corresponding to peak power of the first harmonic, estimated using parabolic interpolation. FF for the rendition was then calculated as the mean FF across time bins for a fixed window defined relative to syllable onset. We similarly excluded introductory notes and call-like syllables, which both consist largely of broadband noise and lack well-defined pitch.

## Electrode array microdrives for neural recordings

Custom-built microdrives, inspired by *Vandecasteele et al., 2012*, were constructed to hold a custom-built array of tungsten electrodes. For one bird we used two 0.5 MOhm and two 6.0 MOhm electrodes in each array organized in a diamond formation (see below for spatial spread). For other birds we used four (diamond) or five (pentagon) 0.5 MOhm electrodes. In all cases, we used tungsten electrodes from Microprobes (WE30010.5F for 0.5 MOhm, and WE30016.0F for 6.0 MOhm). Microdrives consisted of a movable shuttle onto which electrodes were affixed, allowing manual adjustment of the position of electrodes along the *z*-axis with a resolution of 26.5 μm (one-eighth turn of a 00–120 screw). Electrodes were stabilized in the horizontal (*x–y*) plane by passing them through tight-fitting polyimide tubes (0.0056″ ID, 0.00075″ WD) glued to the static parts of the drive. Silver wires (diameter, 0.003″ bare, 0.0055″ teflon-coated) connected each electrode to its pin on an Omnetics connector (A79042-001). Low impedance reference electrodes were made by cutting tungsten electrodes described above to a blunt tip. Electrodes for the two microdrives (LMAN and RA) were wired to different channels on the same connector.

Electrodes were positioned in the array such that their tips were in the same horizontal plane. Electrodes were positioned so that there was greater spread along the anterior–posterior axis (0.4–0.5 mm) than the medial–lateral axis (0.2–0.3 mm) to account for the expectation of greater variability in targeting along the anterior–posterior axis, since position in this axis was further from stereotaxic zero than position in the medial–lateral axis (see below for coordinates).

## Implantation of recording microdrives

Implants were performed in the left hemisphere in the following order within a single surgical session with birds under anesthesia: RA microdrive, LMAN microdrive, reference electrode, shared connector. All wiring of electrodes to connectors was completed before implantation.

The location of RA was confirmed by electrophysiological targeting. Single carbon-fiber electrodes (Kation Scientific, Carbostar-1) were lowered gradually at candidate $x$–$y$ locations to depths where RA was expected. RA was detected based on the presence of its characteristic tonic spiking activity (10–20 Hz). The $x$–$y$ location of the center of RA was determined as where the extent of tonic activity extended over a depth of at least 400 µm; we performed up to three penetrations at different $x$–$y$ locations to determine the best estimate of RA's center, which we found to be 2.05–2.15 mm lateral, 0.04 mm posterior to $Y_0$ (i.e., caudal point of the intersection of the midsagittal and transverse sinuses), 2.75–3.15 mm ventral to the brain surface, with beak angle at 42° [by definition, vertical (beak pointing down) was 0° and horizontal was 90°].

A microdrive was then implanted with electrode tips 1.2 mm above the dorsal edge of RA. Next, a microdrive was implanted over LMAN using stereotaxic coordinates: 1.37 mm lateral, 5.18–5.27 mm anterior, 1.25 mm ventral, at a 50° beak angle. This depth was expected to place the electrode tips ~0.5 mm above LMAN. The reference electrode was implanted directly underneath the skull but with minimal penetration of brain, either over the cerebellum or directly between the LMAN and RA implants. The connector was fixed to the skull over the right hemisphere. Dental cement (Coltene Hygenic) was used to secure all implants to the skull; small holes were made in the upper layer of the skull into which dental cement could flow before curing to increase implant stability. A plastic tube, glued to the dental cement base of the implant and surrounding the implant, was used as a protective cap.

## Electrophysiological recordings

After subjects recovered and were singing (1–2 days), they were tethered and then handled a few times a day for acclimatization. Recording sessions began a week or two after surgery after birds consistently sang within tens of minutes after being handled. Starting immediately after lights were turned on in the morning, we slowly lowered (<20 µm/s) the electrodes from their resting positions toward LMAN and RA. Localization within these nuclei was assessed by evaluating tonic activity (RA), song-locked firing rate modulation (LMAN and RA), and depth. Post hoc histological verification confirmed that recordings were within LMAN and RA (see below). At the end of each session, electrodes were raised to a position with the tip at >300 µm above the dorsal edges of RA and LMAN to minimize potential tissue damage within those areas. Recording sessions were separated by multiple days, and different depths were targeted, with no selection criteria (e.g., firing rate or baseline correlation with pitch) for units that were recorded.

Voltage signals were measured using a homemade lightweight headstage (Intan RHD2132 amplifier chip) and the Intan RHD2000 Amplifier Evaluation System. Signals were amplified, filtered (1–12,000Hz pass band), and multiplexed on the headstage, then stored on hard disk for offline analysis.

A total of seven birds received dual LMAN and RA implants. For three of these birds, we were unable to obtain any appropriate data due to song deterioration after surgery (likely due to damage of HVC, RA, or the HVC–RA tract). For analyses of LMAN and RA activity during baseline singing, we obtained recordings from the remaining four birds (including recordings performed separately in LMAN and RA), resulting in a total of $N = 30$ LMAN and 52 RA multi-unit sites (ranging from 3 to 10 LMAN and 5–20 RA sites per bird). Analysis of LMAN–RA cross-covariance in learning experiments included only concurrent LMAN and RA recordings collected over a day of learning. We excluded one of the four birds, for which we were only able to collect concurrent recordings from LMAN and RA during baseline singing, as signal strength had degraded (to the point where spikes were not detectable) before the rate of songs per day had recovered to a level sufficient for training pitch modification. This resulted in a a total of $N = 38$ pairs of LMAN and RA sites across 11 learning sessions in 3 birds. We varied the target syllable and training direction across the experiments (learning sessions) performed for each bird. Neural analyses excluded the two syllables directly following the target syllable, to avoid potential acute startle or feedback effects that may occur due to WN (*Sakata and Brainard, 2008*; *Sakata and Brainard, 2006*), and syllables that were associated with movement artifacts either during baseline or training renditions ($N = 3.3/10.8$ syllables). In each training session we

recorded 1–2 sites in LMAN and 1–5 sites in RA, and considered all pairs of these LMAN–RA sites for LMAN–RA cross-covariance.

## Spike detection

Spikes were detected using the spike clustering software Wave_clus (*Chaure et al., 2018*) run on MATLAB. Briefly, we detected putative spikes by amplitude threshold crossing (threshold of 2.5–3.5 × SNR, minimum refractory period 0.2 ms), mapped those spikes onto a feature space defined by wavelet coefficients, and then clustered spikes in this feature space. The noise cluster was discarded, and all spike clusters were merged into a single multiunit cluster.

Prior literature suggests that a substantial portion of the spikes we recorded was from excitatory projection neurons. First, for LMAN, the ratio of projection neurons to interneurons has been estimated at around 3:1 (*Bottjer et al., 1998*) to 15:1 (*Livingston and Mooney, 1997*). Further evidence that excitatory neurons are more easily detected than inhibitory ones comes from reports of difficulty finding interneurons in slice preparations (*Boettiger and Doupe, 1998*; *Livingston and Mooney, 1997*). One study reported that LMAN projection neurons verified as projecting to RA via antidromic stimulation have similar spiking activity to unverified ones, suggesting that extracellular recordings in LMAN tend to be dominated by projection neuron activity (*Olveczky et al., 2005*). For RA, the tonic firing we found in the multi-unit activity is characteristic of projection neurons (*Leonardo and Fee, 2005*; *Sober et al., 2008*; *Spiro et al., 1999*). Moreover, other studies isolating single neurons have reported low probability of finding putative interneurons in RA *Leonardo and Fee, 2005*; *Sober et al., 2008*; indeed, in slice preparations the ratio of projection neuron to interneurons in RA was found to be around 30:1 (*Spiro et al., 1999*).

## Analysis of temporal structure of LMAN and RA activity during singing

To assess the temporal structure of activity separately in LMAN and RA, we computed the average firing patterns aligned to song motifs (stereotyped sequences of syllables), as shown in *Figure 1—figure supplement 1*. Motifs were identified by visual inspection of song bout spectrograms ($N = 8$ motifs across 4 birds). For a given motif, the timing of syllable onsets and offsets can vary slightly across renditions and recording sessions. To account for this variation and temporally align activity, we linearly time warped all renditions to a 'reference' motif, constructed with syllables and gaps matching the median value of syllables and gaps for that specific motif across all of its renditions. This alignment was performed by shifting each spike so that its fractional time within its containing segment (i.e., syllable or gap) remained unchanged after time warping (*Figure 1—figure supplement 1A*).

To compute a smoothed firing rate function, activity on each rendition was smoothed by first binning spikes (1 ms) then convolving with a Gaussian kernel (5 ms SD). Rendition-averaged activity was then z-scored to facilitate comparison across recording sites, which may have different firing rates, relative to the mean and SD over the entire motif.

To assess the similarity of LMAN and RA activity patterns, we combined all recordings across sessions into separate LMAN and RA datasets, one for each motif, where activity was time warped, smoothed, and averaged over all sites (without first z-scoring) and then mean-subtracted before computing cross-correlation.

For analysis of average LMAN and RA activity during baseline singing, to assess whether the average time lag of maximum LMAN–RA cross-correlation was significantly different from zero, we performed a Monte Carlo permutation test (*Figure 1—figure supplement 1F*). We computed a null distribution under the null hypothesis that LMAN and RA activity patterns have the same temporal profile relative to song. On each shuffle iteration, we randomized the assignment of rendition-averaged neural activity to brain region. We first generated a dataset consisting of average activity patterns, one for each combination of motif and brain area. On each shuffle iteration, we randomly reassigned the brain area labels; this was done independently for each motif, which ensures that the number of LMAN and RA sites assigned to each motif (and therefore bird) remained unchanged. LMAN–RA cross-correlations were computed with this shuffled dataset for 10,000 random permutations. The probability of finding in the shuffled dataset an absolute time lag equal to or greater than the absolute time lag in the real dataset was taken as the two-sided p-value.

## Normalized cross-covariance

Normalized cross-covariance was computed for each pair of LMAN and RA sites for each syllable's premotor activity (neural activity extracted from 100 to 0 ms preceding syllable onset). This calculation was done separately for baseline and training. For baseline, we used the last half of renditions, to minimize potential drift in recordings from lowering electrodes into position at the start of the session. For training, we used the last quarter of the renditions to take the window of maximal learning.

Normalized cross-covariance was calculated in a similar manner to previous birdsong studies (*Hahnloser et al., 2006*; *Kimpo et al., 2003*). We first calculated the average cross-correlation:

$$Cross\left(\tau\right) = \left\langle \frac{1}{T}\sum_{t=1}^{T} r_L\left(t\right) r_R\left(t+\tau\right) \right\rangle$$

where $r_L\left(t\right), r_R\left(t\right)$ are binned spike counts at time bin $t$ for LMAN and RA, respectively, $T$ is number of time bins in a single rendition, $\tau$ is lag between LMAN and RA data in units of time bins (2.5 ms), and $\langle\rangle$ indicates average over all renditions.

The cross-covariance was computed to estimate the extent to which deviations of firing rates in LMAN and RA from their respective means are associated:

$$
\begin{aligned}
C_{L-R}\left(\tau\right) &= \left\langle \frac{1}{T}\sum_{t=1}^{T}\left[r_L\left(t\right) - \bar{r}_L\left(t\right)\right]\left[r_R\left(t+\tau\right) - \bar{r}_R\left(t+\tau\right)\right] \right\rangle \\
&= \left\langle \frac{1}{T}\sum_{t=1}^{T} r_L\left(t\right) r_R\left(t+\tau\right) - \frac{1}{T}\sum_{t=1}^{T} \bar{r}_L\left(t\right) \bar{r}_R\left(t+\tau\right) \right\rangle
\end{aligned}
$$

where $\bar{r}_L\left(t\right), \bar{r}_R\left(t\right)$ are rendition-averaged firing rates in LMAN and RA.

The second term on the right hand side, $\frac{1}{T}\sum_{t=1}^{T} \bar{r}_L\left(t\right) \bar{r}_R\left(t+\tau\right)$, is the cross-correlation of the average firing rates, since it measures the average similarity of LMAN and RA activity while removing the contribution of shared, within-rendition variation in LMAN and RA. It can be estimated by calculating a shuffled cross-correlation (*Brody, 1999*; *Hahnloser et al., 2006*; *Kimpo et al., 2003*; *Perkel et al., 1967*). The shuffled cross-correlation (or 'shift predictor') is computed using data shuffled such that rendition $n$ for one recording site (either LMAN or RA) is compared to a temporally adjacent rendition (i.e., $n + 1$ or $n − 1$) for the other site:

$$shiftpredictor_{L-R} = \frac{1}{\left(N-1\right)}\sum_{n=1}^{N-1}\frac{1}{T}\sum_{t=1}^{T} r_L^n\left(t\right) r_R^{n+1}\left(t+\tau\right)$$

where $n = 1 \ldots N$ indexes the rendition, and the $L − R$ subscript indicates that LMAN renditions ($n$) are chosen to be the one directly preceding the RA rendition ($n + 1$). We also computed the $R − L$ shift predictor:

$$shiftpredictor_{R-L} = \frac{1}{\left(N-1\right)}\sum_{n=1}^{N-1}\frac{1}{T}\sum_{t=1}^{T} r_R^n\left(t\right) r_L^{n+1}\left(t+\tau\right)$$

The final shift predictor was the average of these two:

$$shiftpredictor = \frac{1}{2}\left(shiftpredictor_{L-R} + shiftpredictor_{R-L}\right)$$

Subtracting the shift predictor from the average cross-correlation gives the cross-covariance:

$$C\left(\tau\right) = \left\langle \frac{1}{T}\sum_{t=1}^{T} r_L\left(t\right) r_R\left(t+\tau\right) - shiftpredictor \right\rangle$$

in units of spikes². To rescale cross-covariance in units more easily comparable across the dataset we normalized cross-covariance relative to the standard deviation of the cross-covariance across all the data points (i.e., each combination of rendition $n$ vs. rendition $n + 1$) in the shuffled dataset. This effectively *z*-scores the cross-covariance relative to the mean and standard deviation of the shuffled distribution:

$$C_{normalized}(\tau) = \frac{C(\tau) - \left\langle c_{shuff}^n(\tau) \right\rangle}{stdev\left(c_{shuff}^n(\tau)\right)}$$

where

$$c_{shuff}^n(\tau) = \frac{1}{T}\sum_{t=1}^{T} r_L^n(t)\, r_R^{n+1}(t+\tau) - shiftpredictor$$

and where *stdev* is the standard deviation over all $n$ (from 1 to $N-1$), for the time bin $\tau$ for the shuffled data:

$$stdev\left(c_{shuff}^n(\tau)\right) = \sqrt{\frac{1}{(N-1)-1}\sum_{n=1}^{N-1}\left(c_{shuff}^n(\tau) - \left\langle c_{shuff}^n(\tau)\right\rangle\right)^2}$$

and $\left\langle c_{shuff}^n(\tau)\right\rangle$ is the mean over renditions $n$ (from 1 to $N-1$). In practice, shuffled renditions can be computed with either LMAN renditions preceding RA renditions or vice versa. We combined both kinds of shuffled renditions into a single set $c_{shuff}^n(\tau)$ when computing $\left\langle c_{shuff}^n(\tau)\right\rangle$ and $stdev\left(c_{shuff}^n(\tau)\right)$.

Cross-correlation functions were first linearly interpolated to 1ms resolution then smoothed with a Gaussian kernel (SD = 5 ms) before converting to normalized cross-covariance. Calculation of cross-correlations were implemented using the MATLAB function *XCORR* (with *SCALEOPT* = 'unbiased'). Normalized cross-covariance was computed separately in 60-ms windows, sliding over 5 ms time-steps, in each syllable's premotor window (such that the earliest window spanned from 100 to 40 ms preceding syllable onset, and the latest window from 60 to 0 ms). The cross-covariance functions over all 60-ms windows were then averaged to generate one cross-covariance function for a given 100 ms premotor window.

To summarize the strength of the LMAN-leading peak in a scalar value, we took the average within a 15-ms window centered at the time lag of peak normalized cross-covariance at baseline across all syllables (3 ms, LMAN leading).

For the analysis, splitting interleaved renditions by pitch into two groups ('Stronger' and 'Weaker' bias), we split renditions by comparing their pitch to the median pitch. If pitch deviated from the median pitch in the adaptive direction (i.e., escaping WN), then the rendition was considered to express stronger behavioral bias; if pitch was in the other direction, then weaker bias. This grouping was performed separately for baseline and WN renditions relative to their respective median pitch values, ensuring that the resulting groups consisted of interleaved renditions. We used the last half of the baseline and training renditions. For the 'Baseline' dataset, the adaptive direction was set the same as that for the 'Trained' dataset so that we could specifically measure learning-related change in the relationship between pitch and neural activity by subtracting baseline from training measurements. Normalized cross-covariance was computed as before, separately for each dataset group. Shuffled renditions were constrained to be only those pairs of renditions overlapping with the renditions included in a given group. For example, if rendition $m$ for LMAN is included in the group, then the corresponding shuffle renditions will be LMAN rendition $m$ vs. RA rendition $m-1$ and LMAN rendition $m$ vs. RA rendition $m+1$. At dataset edges, when $m=1$, then RA rendition $N$ was used instead of $m-1$ (which does not exist), and when $m=N$, RA rendition 1 was used instead of $N+1$ (which does not exist).

## Electrical microstimulation of LMAN

We used a modified version of EvTaf that enabled electrical stimulation to be delivered independently of WN feedback. In order to stimulate at a controlled time relative to the syllable targeted for learning, we detected 'predictor' syllables that consistently preceded the targeted syllable. We then delivered stimulation trains (60 ms duration, 200 Hz bilateral stimulation, centered at 50 ms preceding syllable onset) at a fixed delay from this detection so that stimulation began at a premotor latency prior to the WN trigger time. Stimulation renditions were randomly interleaved with catch renditions in which no stimulation was delivered. We stimulated a randomly interleaved 50% of renditions of the targeted syllable only during specific 1- to 3-hr intervals on specific days at baseline and training (1–3 days).

Custom-made 4-wire Pt/Ir microwire arrays (Microprobe; 25 µm diameter, 400–800 kOhm impedance wires) were surgically implanted bilaterally. The arrays were targeted using stereotaxic coordinates for the center of LMAN. Wires were electrically connected to a male Omnetics connector (A8391-001) that enabled electrical connection to an external lead. The four wires in each array were arranged in a rectangular pattern (250–500 µm separation on the rostro-caudal axis, 250–500 µm separation on the medial–lateral axis) for all but one bird. Wire pairs at the same rostral/caudal or medial/lateral level were separated in depth by 0–250 µm. In one bird, the four wires were laid out in a linear array in which wires were separated by 250 µm along the medio-lateral axis, and neighboring wires were separated by 250 µm in depth. In the rectangular configuration, stimulation was between diagonal wires; in linear configuration, stimulation was between the two inner wires.

After recovery from surgery, microstimulation trains (biphasic pulses, total biphasic pulse duration of 0.4 ms, 200 Hz frequency, 30–100 µA) were delivered to LMAN bilaterally by two separate microstimulators (A-M systems Model 2100). We adjusted various microstimulation parameters to globally disrupt LMAN activity without inducing the large, rapid deflections in pitch previously reported following unilateral LMAN microstimulation (*Kao et al., 2005*). First, we bilaterally stimulated LMAN to induce a more global activity disruption than effected with unilateral stimulation (*Kao et al., 2005*). Second, to perturb a large volume of LMAN rather than a specific area, we passed current between pairs of identical wires placed within LMAN (distance between electrodes ≥350 µm), rather than from a single electrode to ground; We selected pairs of arrays for stimulation which elicited minimal baseline pitch deviations. Finally, we set the amplitude of stimulation at a current level below the threshold at which song stoppages or degradation of syllable structure occurred.

We calculated the time latency from short-duration LMAN microstimulation [three pulses for 10 ms ($N$ = 3) or four pulses for 15 ms ($N$ = 1)] to pitch reversion by comparing randomly interleaved unstimulated and stimulated syllables. For each syllable, we made a continuous measurement of pitch at a millisecond timescale (*Charlesworth et al., 2011*). We then calculated the time-varying pitch difference (or residual) of each stimulated syllable from the mean of the unstimulated syllables. We aligned these residuals according to stimulation onset to obtain the latency to pitch reversion, defined as the duration until the beginning of a 10-ms (or longer) time window in which the residuals were significantly shifted toward baseline pitch for all time bins.

## Post-mortem localization of recording and stimulation sites

For both recording and microstimulation experiments, we marked the location of electrodes by first lesioning brain tissue and then performing histology to map those lesions relative to sites of recording (LMAN and RA) or microstimulation (LMAN) (see *Figure 1—figure supplement 2*). Lesions were performed by passing 100 µA current for 4 s. After lesions, birds were deeply anesthetized and perfused with 4% formaldehyde. Brains were removed and post-fixed for a few hours to overnight. We performed histology on sectioned tissue (40 µm thick, coronal). Electrode tips were localized by identifying lesions and tracts by identifying tissue damage. LMAN and RA were visualized by immunostaining for calcitonin gene-related peptide (Sigma, RRID: AB_259091, 1:5000 to 1:10,000) (*Bottjer et al., 1997*). For microstimulation experiments, we confirmed that lesions were in LMAN. For neural recording experiments, two lesion sites were made, one immediately dorsal and another immediately ventral to LMAN and RA, in order to retain the integrity of tissue within each area for histology. We confirmed in histology that lesions were indeed positioned dorsal and ventral to LMAN and RA, such that electrodes would be expected to be within these regions when at stereotaxic depths used during recordings.

## Statistical tests

The main recording results were analyzed using mixed effects modeling to capture potential hierarchical effects based on experimental session, because a given experimental session may contribute multiple pairs of sites, which are not completely independent; in such cases we modeled responses (changes in LMAN–RA cross-covariance) with fixed effects for the covariate of interest, with random effects for intercept and the covariate of interest grouped by experiment ID.

## Acknowledgements

We thank Alla Karpova, Lena Veit, Michael Berger, Stefan Lemke, Tomoki Suzuki, Eszter Kish, and Brad Colquitt for comments on the manuscript, and other members of the Brainard lab for feedback.

## Additional information

### Funding

| Funder | Grant reference number | Author |
|--------|------------------------|--------|
| Howard Hughes Medical Institute | | Michael S Brainard |

The funders had no role in study design, data collection, and interpretation, or the decision to submit the work for publication.

### Author contributions

Lucas Y Tian, Timothy L Warren, Conceptualization, Data curation, Software, Formal analysis, Investigation, Visualization, Methodology, Writing – original draft, Writing – review and editing; William H Mehaffey, Conceptualization, Software, Methodology, Writing – review and editing; Michael S Brainard, Conceptualization, Resources, Supervision, Funding acquisition, Writing – original draft, Project administration, Writing – review and editing

### Author ORCIDs

Lucas Y Tian ⓘ https://orcid.org/0000-0002-7346-7360
Timothy L Warren ⓘ https://orcid.org/0000-0002-4429-4106
William H Mehaffey ⓘ http://orcid.org/0000-0002-0818-3580
Michael S Brainard ⓘ http://orcid.org/0000-0002-9425-9907

### Ethics

All procedures were performed in accordance with animal care protocols approved by the University of California, San Francisco Institutional Animal Care and Use Committee (IACUC, Approval #: AN185512-02E).

### Decision letter and Author response

Decision letter https://doi.org/10.7554/eLife.83223.sa1
Author response https://doi.org/10.7554/eLife.83223.sa2

## Additional files

### Supplementary files

• MDAR checklist

### Data availability

Data and code to generate figures are deposited in Dryad at https://doi.org/10.5061/dryad.n8pk0p30c.

The following dataset was generated:

| Author(s) | Year | Dataset title | Dataset URL | Database and Identifier |
|-----------|------|---------------|-------------|-------------------------|
| Tian LY, Warren T, Mehaffey W, Brainard M | 2023 | Dynamic top-down biasing implements rapid adaptive changes to individual movements | https://doi.org/10.5061/dryad.n8pk0p30c | Dryad Digital Repository, 10.5061/dryad.n8pk0p30c |

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
