## [Editor Report]

This paper shows that the output of a songbird basal ganglia cortical loop exhibits activity that significantly covaries with a downstream primary motor area with the temporal specificity necessary to promote learning and adaptive premotor bias. As the first study to record from two distant sites at once in singing birds, this study also provides exceptional evidence for temporally precise coordination between two motor areas in the service of vocal learning. The paper provides fundamental implications for understanding the precise neural mechanisms associated with top-down modification of neural commands during motor learning.

---

## [Decision Letter]

**Decision letter after peer review:**

Thank you for submitting your article "Dynamic top-down biasing implements rapid adaptive changes to individual movements" for consideration by *eLife*. Your article has been reviewed by 3 peer reviewers, including Jesse H Goldberg as the Reviewing Editor and Reviewer #1, and the evaluation has been overseen by Michael Frank as the Senior Editor. The following individuals involved in the review of your submission have agreed to reveal their identity: Marc Schmidt (Reviewer #2); Arthur Leblois (Reviewer #3).

Tian et al. impressively record from two motor areas at once in singing birds to test if a premotor cortical area, LMAN, covaries with activity in a primary one, RA, in a way that would support learning. They find that LMAN activity covaries with RA activity at a lag consistent with driving a premotor bias and, moreover, that this covariation is significantly increased in the specific time window of the song where bias is being most strongly driven. Disruptive microstimulation of LMAN in this window reduced learning-associated bias. Though the main results in this paper are consistent with dominant models of birdsong production and learning going back decades (e.g. Kao et al., 2005; Olveczky et al., 2005; Andalman et al., 2008; Charlesworth et al., 2009; Fee and Goldberg, 2011), these results provide methodologically impressive confirmation that LMAN drives RA activity to drive adaptive bias. It's also meaningful that these covariations were strong enough to be picked up by a pair of randomly targeted LMAN and RA sites.

The reviewers were unanimously supportive of this strong paper, and provided several issues that, if addressed, would likely increase the impact and appeal:

1) Song is complex with many varying acoustic parameters, such as amplitude, entropy, and pitch. It is thought that pitch is controlled by only a subset of the syringeal muscles, and also that there is topography in the LMAN-RA-MN-muscle pathway. Thus, one might expect only a small fraction of neurons/sites in the LMAN-RA pathway to be associated with pitch with enough strength that one would pick it up in single unit recordings from order ~100 neurons. Indeed a past study (Sober et al., 2008) found that activity in a small fraction of RA neurons was weakly correlated with pitch variation. So it's really surprising that a pitch-contingent learning paradigm produced significant co-variance changes in the LMAN-RA pathway that could be picked up in the present study. All three reviewers zeroed in on this issue.

Three possible explanations are with consideration. First, one wonders if the authors were recording specifically from pitch-associated sites. Can the authors please elaborate on what fraction of LMAN and RA recording sites in the present study exhibited significant covariance with pitch? Second, if an LMAN-RA recording site pair does not exhibit a significant correlation with pitch but nonetheless exhibits enhanced co-variation in the pre-target window in a pitch shift paradigm, this would support a different interpretation of the results. For example, it's possible that the extent by which LMAN can drive RA is gated by cholinergic inputs from VP, which might signal predicted uncertainty in the song in the precise moment preceding the target time (e.g. Chen et al., 2019; Puzerey et al., 2018). No new experiments are required here, but these possibilities (or other considerations of the mechanisms by which LMAN-RA covariance is temporally gated) could, in the discussion or elsewhere, motivate future studies. (For example, if Ach-mediated predicted uncertainty is the key to promoting LMAN-RA covariance, then photoactivation of Ach inputs to RA at a moment in song might increase LMAN drive and, secondly, non-pitch-contingent DAF that does not drive explicitly learning associated bias would also be sufficient to promote temporally precise increases in LMAN-RA covariance)

Finally, related to these questions, can the authors please check if their recording sites exhibited correlations with pitch (e.g. as in Sober et al., 2008)? Did some channels have stronger pitch correlations than others and how did this affect the cross-covariance on that channel? One possible analysis would be to plot on the x-axis the degree of correlation with pitch and on the y-axis the degree of enhanced cross-covariance at the DAF-targeted time of the song; other analyses may also address this.

1.1. Please elaborate discussion as suggested above.

1.2. Please test if recording sites exhibited correlations with pitch and if this degree of correlation affected the temporal specificity of the peak in cross-covariance relative to DAF-target time, as suggested above. We realize this may be technically difficult and requires the holding of units for a long time (to get enough spikes to syllable renditions for statistical power.) If such a channel-pitch analysis is not possible, please address in the revision what the limitations were. Though all referees supported the clarification/testing of this important issue.

2) Multiple recording sites in both RA and LMAN provide sensible internal controls for the cross-covariance and its increase in the window before bias production in pitch-shift experiments. Can the authors analyze LMAN-LMAN co-variance in the same way to test for LMAN-RA co-variance to examine intra-LMAN activity co-fluctuations? A negative result would support the specificity of the LMAN-RA covariance but a positive result would indicate that within-LMAN dynamics also exhibit interesting learning-related changes.

2.1. Please test for within-LMAN covariations as suggested above.

3) Can the authors please provide more justification for the duration of the premotor window used for covariance calculation? The authors chose to quantify correlation and covariance in a 100ms window preceding the production of the target syllable. They justify this choice by citing previous articles relying on a premotor window or highlighting syllable-related motor activity in the considered brain areas. However, to our knowledge, previous work rather relied on a shorter premotor window (50ms or less). While the difference is not very big, the timescale over which neural activity affects behavior is still only partially understood. Birds can shift the pitch of a syllable with an ms resolution as shown by the same group. And 100ms is beyond the timescale of the fine gesture underlying song production. Therefore, it is not clear why this choice is made and whether a shorter window would provide similar results.

3.1. Test if analyses hold for a shorter window (e.g. 50 ms – this analysis could go in supplement) or, alternatively, the authors could better justify their choice of a longer premotor window in light of this comment.

---

## [Author Response]

Essential revisions:1) Song is complex with many varying acoustic parameters, such as amplitude, entropy, and pitch. It is thought that pitch is controlled by only a subset of the syringeal muscles, and also that there is topography in the LMAN-RA-MN-muscle pathway. Thus, one might expect only a small fraction of neurons/sites in the LMAN-RA pathway to be associated with pitch with enough strength that one would pick it up in single unit recordings from order ~100 neurons. Indeed a past study (Sober et al., 2008) found that activity in a small fraction of RA neurons was weakly correlated with pitch variation. So it's really surprising that a pitch-contingent learning paradigm produced significant co-variance changes in the LMAN-RA pathway that could be picked up in the present study. All three reviewers zeroed in on this issue.Three possible explanations are with consideration. First, one wonders if the authors were recording specifically from pitch-associated sites. Can the authors please elaborate on what fraction of LMAN and RA recording sites in the present study exhibited significant covariance with pitch? Second, if an LMAN-RA recording site pair does not exhibit a significant correlation with pitch but nonetheless exhibits enhanced co-variation in the pre-target window in a pitch shift paradigm, this would support a different interpretation of the results. For example, it's possible that the extent by which LMAN can drive RA is gated by cholinergic inputs from VP, which might signal predicted uncertainty in the song in the precise moment preceding the target time (e.g. Chen et al., 2019; Puzerey et al., 2018). No new experiments are required here, but these possibilities (or other considerations of the mechanisms by which LMAN-RA covariance is temporally gated) could, in the discussion or elsewhere, motivate future studies. (For example, if Ach-mediated predicted uncertainty is the key to promoting LMAN-RA covariance, then photoactivation of Ach inputs to RA at a moment in song might increase LMAN drive and, secondly, non-pitch-contingent DAF that does not drive explicitly learning associated bias would also be sufficient to promote temporally precise increases in LMAN-RA covariance)Finally, related to these questions, can the authors please check if their recording sites exhibited correlations with pitch (e.g. as in Sober et al., 2008)? Did some channels have stronger pitch correlations than others and how did this affect the cross-covariance on that channel? One possible analysis would be to plot on the x-axis the degree of correlation with pitch and on the y-axis the degree of enhanced cross-covariance at the DAF-targeted time of the song; other analyses may also address this.1.1. Please elaborate discussion as suggested above.

We have noted in the discussion (and methods) that recording sites were not specifically selected for having activity correlated with pitch. We agree with the reviewers that it is striking that increases in cross-covariance were observed in a distributed fashion across randomly selected recording sites. We have elaborated in the discussion the implication that changes to activity associated with modification of target syllable are therefore likely to be distributed, and discuss how this might be consistent with our prior understanding of anatomy and physiology of RA (lines 269 – 276):

“That increases in cross-covariance can be observed at arbitrarily selected recording locations in RA and LMAN suggests that the associated changes in these areas are distributed across neurons. At least for RA, the possibility that activity changes are correlated on a rendition-by-rendition basis across neurons is supported both by the known architecture of RA [in which the inhibitory inputs to multiple projection neurons are highly correlated (Miller et al., 2017; Spiro et al., 1999)] and by an understanding that correlated activity of RA projection neurons supports the ability of these neurons to drive downstream structures and influence song (Sober et al., 2008).”

We further have expanded the discussion to discuss alternative mechanisms for the enhancement of LMAN-RA cross-covariance, including gating mechanisms (lines 277 – 284):

“An alternative model is that top-down bias is implemented by altering the efficacy of LMAN synapses in RA—potentially in the absence of changes within LMAN—either by plasticity at these synapses, or by changes to neuromodulatory inputs to RA that act to alter the gain of an appropriate set of LMAN synapses in RA. Indeed, prior work has supported a role for noradrenergic (Sheldon et al., 2020; Solis and Perkel, 2006) and cholinergic (Puzerey et al., 2018) inputs to RA in modulating song. Overall, distinguishing between possible mechanisms for learning-related changes to LMAN’s top-down bias will require elucidation by studies monitoring larger numbers of neurons during learning.”

1.2. Please test if recording sites exhibited correlations with pitch and if this degree of correlation affected the temporal specificity of the peak in cross-covariance relative to DAF-target time, as suggested above. We realize this may be technically difficult and requires the holding of units for a long time (to get enough spikes to syllable renditions for statistical power.) If such a channel-pitch analysis is not possible, please address in the revision what the limitations were. Though all referees supported the clarification/testing of this important issue.

We have performed this analysis and found no evidence for the hypothesized relationship. We first measured the correlation between baseline firing rate and pitch for each recording site in LMAN and RA. Consistent with Sober (2008), we found a trend towards more positive than negative correlations. We then asked whether the strength of correlations in LMAN or RA predicted the change in cross-covariance between LMAN and RA during learning (see Figure 2—supplement 2 in revised manuscript). In our data set, we did not find a significant relationship between the strength of either LMAN or RA correlations and the magnitude of increase in cross-covariance. However, due to the small sample size (sites and renditions during baseline singing), we do not think that these negative findings speak strongly against the possibility that such a relationship would be detected with a larger data set. As described above, we also further discuss in the revision possible mechanisms that could contribute to increased cross-covariance.

2) Multiple recording sites in both RA and LMAN provide sensible internal controls for the cross-covariance and its increase in the window before bias production in pitch-shift experiments. Can the authors analyze LMAN-LMAN co-variance in the same way to test for LMAN-RA co-variance to examine intra-LMAN activity co-fluctuations? A negative result would support the specificity of the LMAN-RA covariance but a positive result would indicate that within-LMAN dynamics also exhibit interesting learning-related changes.2.1. Please test for within-LMAN covariations as suggested above.

We agree that this would be an informative control analysis; however, in our dataset, we do not have enough data to adequately perform this analysis. We had four sessions with two LMAN sites and no sessions with more than two sites, yielding only four pairs of LMAN sites for analysis.

3) Can the authors please provide more justification for the duration of the premotor window used for covariance calculation? The authors chose to quantify correlation and covariance in a 100ms window preceding the production of the target syllable. They justify this choice by citing previous articles relying on a premotor window or highlighting syllable-related motor activity in the considered brain areas. However, to our knowledge, previous work rather relied on a shorter premotor window (50ms or less). While the difference is not very big, the timescale over which neural activity affects behavior is still only partially understood. Birds can shift the pitch of a syllable with an ms resolution as shown by the same group. And 100ms is beyond the timescale of the fine gesture underlying song production. Therefore, it is not clear why this choice is made and whether a shorter window would provide similar results.3.1. Test if analyses hold for a shorter window (e.g. 50 ms – this analysis could go in supplement) or, alternatively, the authors could better justify their choice of a longer premotor window in light of this comment.

We chose this premotor window based on previous microstimulation-based estimates of the latency between song system neural activity and behavior. For RA, latencies have been estimated at ~15 ms (Fee 2004), ~55 ms (see example experiment in Figure 7A in Vu et al., 1994), and ~70 ms (Ashmore et al., 2005; this value is relatively high because it reflects the latency from stimulation until song recovery). For LMAN, latencies have been estimated at 35 to 70 ms (Kao et al., 2005), 20 to 42 ms (Giret et al., 2014), and 10 to 60 ms (Kojima et al., 2018). These values are consistent with latency estimates based on rendition-by-rendition covariation between RA activity and song acoustic features [~25 ms (Wohlgemuth et al., 2010) and ~40 ms (Sober et al., 2008)]. Given this variation in estimated latencies between LMAN/RA activity and behavior, we sought to maximize our chances of detecting learning-related changes to LMAN-RA cross-covariance by using a 100ms window (ending at syllable onset). To clarify this rationale, we now note in the revisions that this premotor window is *centered* at 50ms before syllable onset – a value that is within the range of premotor latencies reported above. At the reviewers’ suggestion, we have also repeated calculations with a shorter window (60 ms, centered at 30 ms before syllable onset), which leads to qualitatively similar results compared to the original 100 ms window. This was verified for the three main LMAN-RA cross-covariance results from the paper, and is now reported in the following figures in the revised manuscript:

Figure 2—Figure Supplement 1A.

Figure 3—Figure Supplement 1A.

Figure 4—Figure Supplement 1.